

# Urban-rural patterns and driving factors of particulate
# matter pollution decrease in eastern china
Zhihao Song1, 2, Bin Chen1, 2
1 College of Atmospheric Sciences, Lanzhou University, Lanzhou 730000, China
2 Institute of Meteorological Artificial Intelligence Research, Lanzhou University, Lanzhou 730000, China
*Correspondence to*: Bin Chen (chenbin@lzu.edu.cn)
**Abstract.** Urban-rural patterns of particulate matter (PM) pollution reduction in China remain poorly
understood. Using an interpretable end-to-end machine learning model framework from original satellite
data, we identified changes in urban and rural PM pollution and the underlying drivers. During the period
2015-2023, the average decrease rates of PM10 and PM2.5 in eastern China were -4.1±1.1 µg/m³/month
and -2.4±0.8 µg/m³/month, respectively. The rate of decrease in urban areas was higher than that in rural
areas, which played a dominant role in PM reduction. Significant reductions in PM concentrations were
observed in urban core areas, suburbs, towns and regions with high agricultural pressure. The
interpretability analysis showed that temperature and interannual variability were the main drivers of PM
pollution reduction. However, only interannual variability showed a significant decreasing trend in its
effect on PM pollution, while other driving factors showed periodic variations. Furthermore, there were
differences in the drivers of PM reduction between urban and rural areas, particularly with interannual
variability in particular contributing to PM pollution reduction in urban areas, but having a lesser impact
in most rural areas. This study reveals the urban-rural patterns of PM pollution reduction in eastern China,
and highlights the need for differentiated air pollution control strategies in urban and rural areas.
**1 Introduction**
Air pollution caused by $PM_{2.5}$ and $PM_{10}$ (airborne particulate matter with diameters less than 2.5µm
and 10µm, respectively) has adversely affected China's atmospheric environment (Huang et al., 2014a;
Zhang et al., 2012). PM pollution is now considered the greatest environmental risk factor for global
human health (Apte et al., 2015), as exposure to PM can trigger various respiratory and cardiovascular
diseases (Burnett Richard et al., 2014; West et al., 2016; Cohen et al., 2017). The indirect health risks
associated with PM exposure (Yin et al., 2020) contribute to millions of premature deaths annually in



China (Burnett et al., 2018). To mitigate the escalating risks of particulate matter exposure and reduce
the public health burden, the Chinese government introduced the "Air Pollution Prevention and Control
Action Plan" in 2013 (State Council of the People's Republic of China, 2013). This initiative aims to
implement policies to improve energy efficiency, reduce energy-related pollution, and curb
anthropogenic emissions to control particulate matter pollution in the atmosphere (State Council of the
People's Republic of China, 2014). As a result of this initiative, China's atmospheric particulate matter
pollution has improved significantly (Cheng et al., 2021). Between 2013 and 2017, the annual average
concentration of PM2.5 decreased by 28-40% (Zheng et al., 2018; Ministry of Ecology and Environment
of the People's Republic of China, 2017), and the population-weighted national annual average
concentration of PM2.5 decreased by 32% (Xue et al., 2019). Data from the National Air Quality
Monitoring Network show that between 2013 and 2020, the annual average PM2.5 concentration in urban
areas of China decreased from 72 µg/m³ to 33 µg/m³ (Song et al., 2023). As a result, the Clean Air Action
has achieved remarkable results in reducing PM pollution (Zhang et al., 2019b).

It is widely accepted that improvements in air quality can be attributed to both reductions in

anthropogenic emissions (Geng et al., 2019; Zheng et al., 2023; Zhao et al., 2018) and changes in
meteorological conditions (An et al., 2019; Cao and Yin, 2020; Chen et al., 2020a). To assess the driving
factors behind changes in PM concentration trends, it is essential to distinguish between anthropogenic
emissions and meteorological factors (Zhong et al., 2018). Zhong et al. (2021) found that PM2.5
concentrations decreased by 44% from 2013 to 2019, and by 34% when the influence of meteorological
conditions was excluded, thus demonstrating the effectiveness of emission reduction measures. Qiu et al.
(2022) used the GEOS-Chem chemical transport model to simulate the impact of anthropogenic
emissions on PM pollution trends and provided recommendations for attributing PM pollution trends to
emission changes. Vu et al. (2019) used machine learning to assess the impact of air quality trends in
Beijing and found that PM2.5 and PM10 concentrations decreased by 34% and 24%, respectively, after
excluding meteorological influences, attributing the decrease to reduced coal burning. Zhai et al. (2019)
used a stepwise multiple linear regression (MLR) model to quantify PM2.5 trends in China between 2013
and 2018, and found that meteorological conditions contributed about 12%. However, Xiao et al. (2021)
used statistical methods to separate the contributions of emissions and meteorology to long-term PM2.5
trends in East China, and found that meteorological contributions were even higher in certain years.



Overall, distinguishing the contributions of anthropogenic emissions and meteorological changes to PM
pollution is crucial to improve understanding of pollution processes and to inform pollution control
policies and future air quality predictions.
However, the urban-rural patterns of PM pollution improvement remain poorly understood in
existing research (Chen et al., 2020b). Many studies on PM pollution either focus on highly polluted
regions (such as the Beijing-Tianjin-Hebei region) (Chen et al., 2019a; Chen et al., 2019b), or on
developed regions with a high concentration of large cities (such as the Yangtze River Delta and the
Pearl River Delta) (Gui et al., 2019; He et al., 2017). This focus is mainly due to the high concentrations
of air pollutants in developed cities (Sicard et al., 2023), where PM pollution poses a significant public
health threat to densely populated urban areas (Brauer et al., 2016; Southerland et al., 2022). Although
PM pollution in urban areas highlights the importance of environmental governance, rural areas, with
different consumption habits and living conditions (e.g., solid fuel burning in households) (Li et al.,
2014)), may experience air pollution that differs from urban areas (Wang et al., 2024a). In certain seasons
and regions, PM exposure factors in rural areas are generally higher than those in urban areas, with
exposure levels reaching up to 70% (Wang et al., 2024b). Therefore, the contribution of these regions to
PM pollution improvement may differ (Li et al., 2024b). Without targeted assessments, perceptions of
the relative importance of urban and rural areas in China's air pollution control efforts may be distorted,
hindering the development of appropriate environmental policies and the promotion of green
development in urban and rural construction (Yang et al., 2024).
This study advances the understanding of the current status and driving factors of urban-rural PM
pollution improvement using interpretable machine learning methods. First, by integrating satellite
observation data, meteorological data, and geographic information, we use a multiple-output extreme
trees (MOET) model to capture the spatiotemporal distribution of PM (including PM10 and PM2.5)
across China and assess the patterns of PM pollution improvement. We then use various machine learning
interpretability techniques, such as relative importance, tree interpreters, and SHAP values, to quantify
the contributions of anthropogenic emissions and meteorological changes to PM pollution improvement.
To investigate potential differences in the results between urban and rural areas, we use land use data to
distinguish urban from rural regions in eastern China. This study aims to address the following three
questions: (1) What are the spatio-temporal patterns of PM pollution improvement in urban and rural



areas of China? (2) What are the main driving factors behind the differences in PM pollution
improvement between urban and rural areas? (3) What are the specific contributions of each driving
factor to PM pollution improvement? Answering these questions is crucial for a comprehensive
understanding of the dynamics of urban and rural atmospheric particulate pollution control in China.
**2 Data and Methods**
**2.1 Satellite TOAR data and ground-based PM observations**
Previous studies have shown that satellite-observed top-of-atmosphere reflectance (TOAR) data
can be used to estimate near-surface air pollutants (Chen et al., 2024a; Yang et al., 2023; Song et al.,
2024). In particular, the TOAR data from the Himawari-8 satellite have demonstrated excellent
performance in pollutant estimation (Hu et al., 2022; Liu et al., 2019). The Advanced Himawari Imager
(AHI) on board the Himawari-8 satellite is an advanced passive observation instrument with 16
observation channels, providing a spatiotemporal resolution of up to 10 minutes and 0.5 km (Bessho et
al., 2016). Based on the sensitivity of the AHI sensor (Yoshida et al., 2018), three visible channels (0.46
μm, 0.51 μm, and 0.64 μm) and two near-infrared channels (0.86 μm and 2.3 μm) were used in this study.
TOAR data from the AHI imager were obtained from the Himawari Monitor P-Tree System data
download website of the Japan Meteorological Agency (https://www.eorc.jaxa.jp/ptree/index.html).
The ground-based PM data were provided by the China National Environmental Monitoring Center
(CEMC) (http://www.cnemc.cn) and were calibrated and quality controlled according to the Chinese
National Standard GB 3095-2012 (Ministry of Ecology and Environment of the People's Republic of
China, 2012). In this study, hourly mean PM10 and PM2.5 data were collected from approximately 1,400
stations in eastern China (102-136°E, 16-56°N) for the period from 1 September 2015 to 31 August 2023.
Observations with PM2.5 concentrations above 600 μg/m³ or PM10 concentrations above 1,000 μg/m³,
as well as those with concentrations below 1 μg/m³, were excluded (Shi et al., 2024).
**2.2 Meteorological data and geographic information data**
Studies assessing the impact of meteorological factors on PM pollution have identified temperature,
humidity, and wind as the main variables influencing PM2.5 concentrations, with their effects
significantly outweighing those of other factors. Among these, temperature has the most significant and




stable influence (Chen et al., 2018b). In this study, meteorological data were obtained from the ERA-5
reanalysis dataset provided by the European Centre for Medium-Range Weather Forecasts
(https://cds.climate.copernicus.eu/cdsapp#!/dataset/). The dataset includes boundary layer height (BLH),
relative humidity (RH), surface pressure (SP), 2-metre air temperature (T2M), wind direction (WD),
wind speed (WS), and net solar radiation at the surface (NSR), with spatial resolutions of $0.1° \times 0.1°$ or
$0.25° \times 0.25°$ (Hersbach et al., 2020). Geographic information can also influence pollutant concentrations
to some extent due to variations in meteorological conditions (Chen et al., 2018a; Chen et al., 2021). The
geographic information data used in this study include elevation (HEIGHT), land cover type (LUCC),
and population density (RK).

**2.3 Data integration and development of the Multiple-Output Extreme Trees Model**

The resolution of the meteorological and geographic information data was adjusted to $0.05° \times 0.05°$

using bilinear interpolation. All data were then matched with station data according to the $0.05° \times 0.05°$
grid of the Himawari-8 satellite. The specific matching method is described in detail in Chen et al. (2022c)
and Song et al. (2022b).

The DOET model is developed on the basis of the Extreme Trees (ET) model (Geurts et al., 2006),

which is capable of simultaneously handle multi-target variable output tasks. The ET model is similar to
the Random Forest (RF) model, both of which consist of multiple decision trees. However, whereas the
RF model randomly samples data with replacement, the ET model uses all available samples. After
determining the samples and features, the ET model constructs decision trees based on optimal partition
attributes. This process is repeated until a sufficient number of decision trees have been constructed to
form the ET model. Finally, the average regression results of all decision trees in the ET are used as the
final output. Several studies have confirmed that the ET model has excellent fitting performance (Qin et
al., 2020; Zhang et al., 2022a; Chen et al., 2022a).

In this study, three model parameters were optimized: the number of trees (n_estimators), the

maximum depth of the model (max_depth), and the minimum number of samples required to split a node
(min_samples_split). After balancing the accuracy and efficiency of the model, these parameters were
set to 70, 100, and 5, respectively. The model, which uses satellite observations, meteorological data,
and geographical information to estimate near-surface PM concentrations, can be expressed as:



$(PM_{10}, PM_{2.5})$
$= f\big(TOAR_{1,2,3,4,6}, BLH, RH, SP, T2M, WD, WS, Height, LUCC, RK, year, mon, doy, hour\big)$    (1)
Here, $f$ represents the DOET model, and $TOAR_{1,2,3,4,6}$ denotes the radiance values of the three
visible channels (0.46 μm, 0.51 μm, and 0.64 μm) and the two near-infrared channels (0.86 μm and 2.3
μm). $BLH, RH, SP, T2M, WD$ and $WS$ are meteorological variables, while $Height, LUCC$ and $RK$
represent geographical information. The variables year, mon (month), doy (day of the year), and hour
are temporal information reflecting the influence of anthropogenic emissions on PM pollution (Wei et
al., 2020). Specifically, year and month (mon) are used to represent the interannual and intra-annual
variations in anthropogenic emissions, respectively (Zhang et al., 2019a; Park et al., 2019). The
estimation workflow is illustrated in Figure 1.

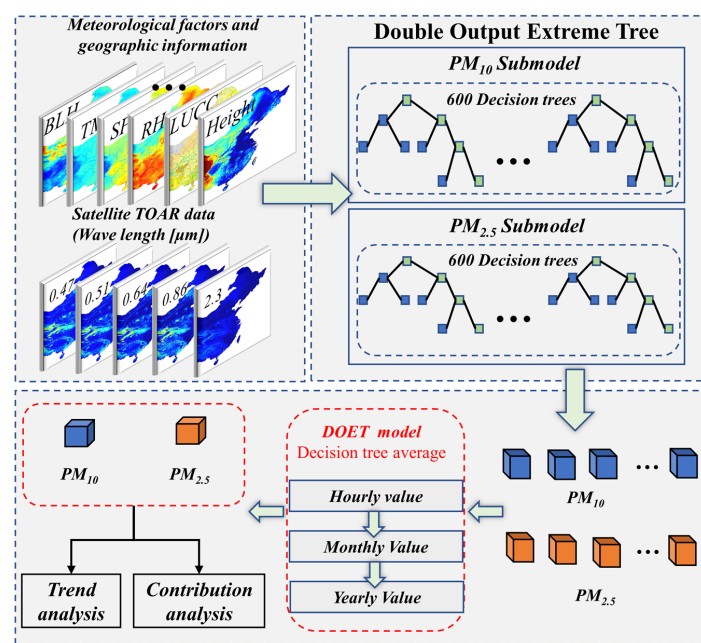

**Figure 1. Workflow of PM data estimation and pollution driving factors assessment.**
Model performance was evaluated using 10-fold cross-validation (Rodriguez et al., 2010),
incorporating sample-based, space-based, and time-based validation methods (Wei et al., 2019).
Evaluation metrics used included the coefficient of determination (R²), root mean square error (RMSE),
and mean absolute error (MAE) for both PM10 and PM2.5 (Chen et al., 2023).
$R^2 = 1 - \frac{ss_{res}}{ss_{tot}}$                                        (2)





$MAE = \frac{1}{n}\sum_{i=1}^{n}|\hat{y}_i - y_i|$                                          (3)
$RMSE = \sqrt{\frac{1}{n}\sum_{i=1}^{n}(\hat{y}_i - y_i)^2}$                                 (4)
**2.4 Machine learning interpretability variables**
To investigate the influence of potential driving factors on PM pollution improvement in eastern
China, we employed relative importance (Berner et al., 2020), tree interpreter (Wang et al., 2022b), and
SHAP values (Lundberg and Lee, 2017) to distinguish the contributions of meteorological changes and
anthropogenic emissions to PM pollution improvement. Relative importance was assessed using the
permutation importance value of the DOET model, defined as the average reduction in model accuracy
when a single feature value is randomly shuffled (Yang et al., 2022).
The permutation importance of each variable was calculated using the "permutation_importance"
library in Python. To reduce uncertainty, the training process was repeated 20 times for each grid point
to obtain robust estimates of relative importance. The tree interpreter was applied using the
'tree_interp_functions' library in Python, which is designed for predictions based on decision tree
ensemble models and facilitates the decomposition of each prediction into bias and feature contribution
components. ([https://github.com/andosa/treeinterpreter/tree/master](https://github.com/andosa/treeinterpreter/tree/master)).
SHAP values are based on Shapley value theory, which explains model predictions by calculating
the relative contribution of each feature to the output (He et al., 2024). These values reflect not only the
influence of features on individual samples but also indicate the positive and negative contributions of
these influences. SHAP explanations can be applied to any machine learning model, including neural
networks and ensemble models, and provide comprehensive and accurate interpretability results. Thus,
the SHAP method provides superior explanations for both local and global model effects (Liu et al., 2023;
Hou et al., 2022). In Python, "tree_SHAP" is specifically tailored for decision tree-based machine
learning models, such as the Extreme Tree model, to provide greater accuracy and faster computation.
The interpretability variables described above were applied to the monthly averaged $PM_{10}$ and $PM_{2.5}$
datasets generated by the DOET model.
**2.5 Land cover type classification**
Zhang et al. (2022b) proposed a method to differentiate urban and rural areas based on the gradient





of human land use pressure. In this study, the MCD12Q1 land cover map, with a spatial resolution of 500
meters was used. For grids measuring 5×5 km, urban and rural classifications were determined by the
coverage of specific land cover categories (e.g., urban land and cropland), which reflect the transition
from urban to rural areas and correspond to different levels of human activity. As shown in Table 1 and
Figure S1, urban areas in this study include both urban core areas and suburban regions, while rural areas
are categorized into six types: towns, high agricultural pressure areas, low agricultural pressure areas,
forests and grasslands.
**Table 1. Definitions of urban and rural land cover classes**

| Urban-Rural Land Cover Class | Definition |
|---|---|
| Urban | 50%<Urban grid |
| Suburban | 25%<Urban grid<50% |
| Towns | 12.5%<Urban grid<25% |
| High Agricultural Pressure Areas | 50%<Cropland grid |
| Low Agricultural Pressure Areas | 12.5%< Cropland grid grid<50% |
| Forests | 50%<Forest grid |
| Grasslands | 50%<Grassland grid |
| Other | Remaining unclassified grids (e.g., desert or tundra) |

**3 Results**
**3.1 PM estimation model performance and PM distribution characteristics**
For the period from September 2015 to August 2023 in eastern China, a total of 6,772,429 samples
were matched. After parameter optimization and feature training, the optimal DOET model was derived,
and long-term time-series spatial distribution products for $PM_{10}$ and $PM_{2.5}$ in eastern China were
generated. Figure 2 shows the results of 10-fold cross-validation based on sample, spatial and temporal
validations. Overall, the DOET model showed a high level of accuracy in the estimation of PM data. The
sample-based 10-fold cross-validation results (Figure 2C and 2F) yielded an $R^2$ of 0.87, with RMSE
(MAE) values of 25.82 (14.87) μg/m³ for $PM_{10}$ and 14.36 (8.44) μg/m³ for $PM_{2.5}$. The slope of the fitting
line between observed and estimated values was 0.84. The performance of the DOET model in this study
is comparable to that reported in other studies that estimated PM using Himawari-8 TOAR data (Wang
et al., 2021; Chen et al., 2024b; Yin et al., 2021).



The 10-fold cross-validation results based on spatial and temporal validation were slightly lower
than those based on samples (Figures 2D-E and 2G-H). Spatial validation assessed the performance of
the model in estimating PM concentrations in areas without monitoring stations, after training the model
with samples from areas with stations. Temporal validation involved training the model with samples
from specific years and testing it with data from years not used in training. For these two validation
methods, the R² values for PM$_{10}$ were 0.83 and 0.41, with RMSE values of 29.99 μg/m³ and 55.44 μg/m³,
respectively. For PM$_{2.5}$, the R² values were 0.83 and 0.51, with RMSE values of 16.46 μg/m³ and 28.11
μg/m³, respectively. The results of the sample-based, spatial, and temporal validation indicate that the
proposed DOET model exhibits robust stability.

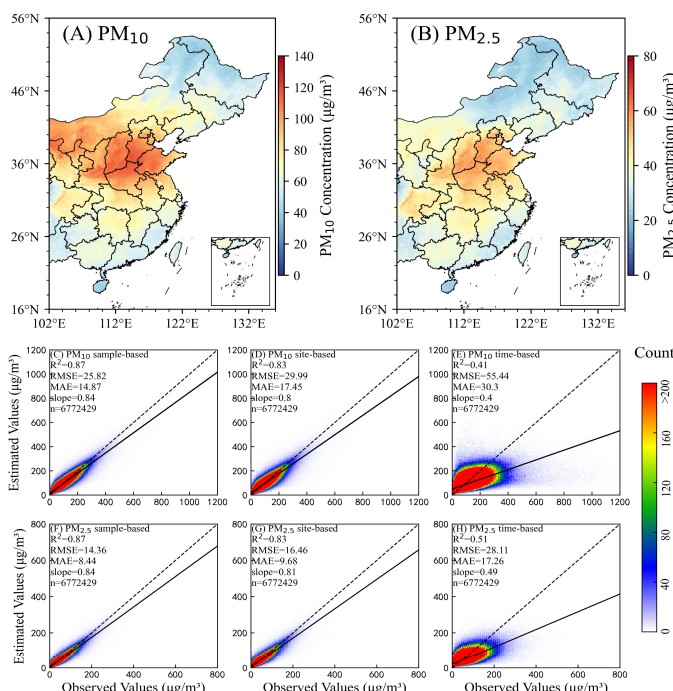


**Figure 2. Spatial distribution of PM$_{10}$ and PM$_{2.5}$ and cross validation results of the DOET model. The dashed**
**lines represent the 1:1 line, while the solid lines show the fitted line between observed and estimated values.**
By inputting TOAR, meteorological elements and geographical information into the optimally
parameterized DOET model, a pollutant estimation dataset for eastern China was generated for the period
September 2015 to August 2023. Due to the incomplete spatial coverage of TOAR data in different
months and hours (Song et al., 2024), the study first calculated monthly averages, which were then used
to derive annual averages. This step helps to minimize errors due to insufficient spatial coverage of the





samples (Ding et al., 2024). As shown in Figures 2A and 2B, the Beijing-Tianjin-Hebei region, the
Sichuan Basin, the Guanzhong region, and central China are hotspots for $PM_{10}$ and $PM_{2.5}$ pollution (Wei
et al., 2021), with concentrations reaching up to 100 μg/m³ for $PM_{10}$ and 60 μg/m³ for $PM_{2.5}$. In addition,
the Inner Mongolia region and northern Gansu, which are frequently affected by dust storms, are also
characterized by high PM10 concentrations (Li et al., 2012). O Overall, the $PM_{10}$ and $PM_{2.5}$
concentrations generated by the DOET model accurately reflect the spatial distribution characteristics of
PM in eastern China, and the estimation results are consistent with those of previous studies (Yang et al.,
2023; Chen et al., 2022b; Song et al., 2022a).

**3.2 Urban-rural differences in PM pollution trends in recent years**

The spatial distribution characteristics of $PM_{10}$ and $PM_{2.5}$ trends from 2015 to 2023 were analysed,
and the results (Figures 3C-F) show a remarkable improvement of PM pollution in eastern China, as
indicated by a significant decreasing trend in PM concentrations. The average decrease for $PM_{10}$ was -
4.1±1.1 μg/m³/month, while for $PM_{2.5}$, it was -2.4±0.8 μg/m³/month. However, this widespread decrease
in PM concentrations showed considerable spatial heterogeneity between urban and rural areas. The
urban and rural decrease trends for $PM_{10}$ were -5.2±1.7 μg/m³/month and -4.1±1.1 μg/m³/month,
respectively, while for $PM_{2.5}$, they were -3.6±1.1 μg/m³/month and -2.3±0.8 μg/m³/month, respectively.
This suggests that the decrease in PM concentrations in rural areas was close to the regional average in
eastern China, while the decrease in urban areas was more pronounced than the overall trend.
From a broader perspective of the changes in particulate matter concentrations in eastern China, the
urban decrease trends for $PM_{10}$ and $PM_{2.5}$ were -0.47 μg/m³/month and -0.33 μg/m³/month, respectively,
while the rural decrease trends were -0.37 μg/m³/month and -0.22 μg/m³/month, respectively. These
results indicate that the reduction trend in rural areas was slower than in urban areas. By 2023, particulate
matter concentrations in urban areas had decreased from about 20 μg/m³ higher than in rural areas to
levels almost equal to those in rural areas.
Urban and rural areas, categorized by land cover type, comprised eight different categories. The
study assessed their respective roles in PM concentration reduction trends and found that all eight
categories showed declining PM trends. However, the regions with the highest PM reduction trends were
mainly four types: urban core areas, suburbs, towns and agricultural land 1 (high agricultural pressure).
In contrast, the reduction trends were less pronounced in agricultural land 2 (low agricultural pressure),



forests, grassland and other areas.

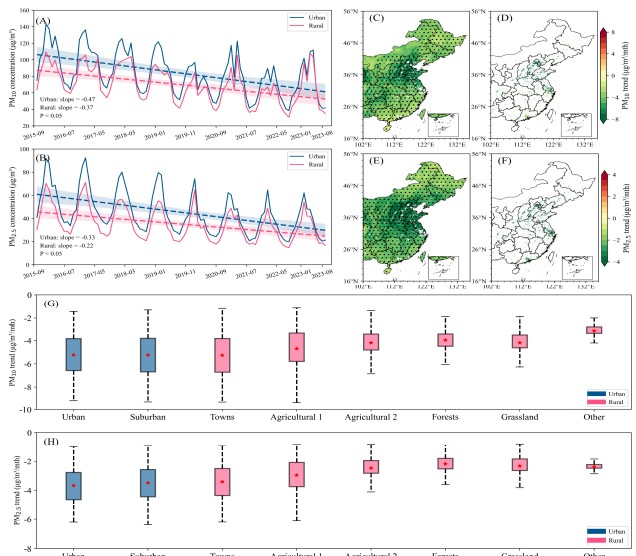


**Figure 3. Analysis of PM concentration trends in eastern China from September 2015 to August 2023. Panels**
**A, C, D, and G represent PM$_{10}$, while panels B, E, F, and H represent PM$_{2.5}$. In the legends of panels G-H,**
**blue indicates urban areas, and red indicates rural areas.**
The trends in PM$_{10}$ and PM$_{2.5}$ concentrations were categorized into four levels based on percentiles:
slow decline (grid points with a decline trend below the 25th percentile), moderate decline (grid points
with a decline trend between the 25th and 75th percentiles), rapid decline (grid points with a decline
trend between the 75th and 95th percentiles), and sharp decline (grid points with a decline trend above
the 95th percentile). As shown in Figure 4, the regions with the most significant changes in urban and
rural PM trends are mainly concentrated in the Beijing-Tianjin-Hebei region, the Guanzhong region and
Central China.
In areas with slow and moderate declines, forests and grasslands accounted for the highest
proportions, ranging from 20.5% to 31.5% and 27.7% to 36.5%, respectively, followed by the first and
second types of agricultural land, which accounted for about 20%. In regions with rapid decline, the first
type of agricultural land had the highest proportion, ranging from 30 to 40%. Urban core, suburban and
rural areas had higher proportions in the fast decline regions, accounting for 6.7%, 7.0% and 8.8% of the
PM10 decline trends and 9.5%, 7.5% and 8.8% of the PM2.5 decline trends respectively. In particular,
the first type of agricultural land had the largest share in the strong decrease regions.



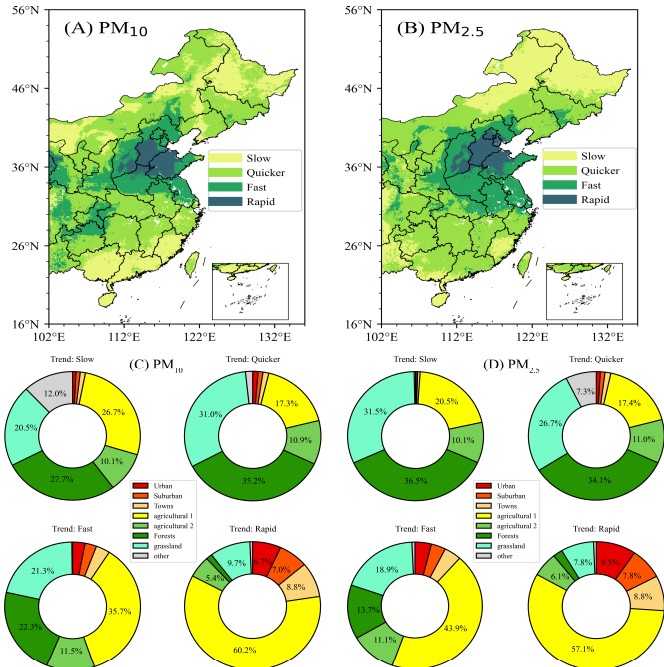

**Figure 4. Spatial distribution of particulate matter trend percentiles and pie charts. The individual color scales in the figure represent different areas.**

**3.3 Assessing potential driving factors for PM pollution improvement and quantifying their contributions**

A DOET model based on monthly PM data was developed to identify the key drivers of urban and rural greening changes in China. Monthly mean $PM_{10}$ and $PM_{2.5}$ concentrations were correlated with meteorological factors and two temporal variables (year and month) representing the effects of meteorological changes and anthropogenic influences, respectively (see Methods for details). The model was cross-validated using a random training set (70%) and a validation set (30%). As shown in Figure S2, the DOET model explains more than 60% of the $PM_{10}$ trends and 80% of the $PM_{2.5}$ trends in eastern China.

The relative importance of each variable in the DOET model was determined using the permutation_importance library. Inter-annual variability, intra-annual variability, air pressure and temperature were identified as significant contributors to the improvement of urban and rural PM pollution in eastern China (relative importance > 10%). Among them, interannual variability was the most influential factor for $PM_{10}$ (28.3±12%), followed by temperature (21.1±15%) (Figure 5A). In



contrast, for PM$_{2.5}$, interannual variability ranked second (32±13.2%), while temperature had a stronger
effect (>40%) (Figure 5B). The spatial distribution of the relative importance of the four main
contributing factors, shown in Figures 5C-R, indicates that regions with high relative importance values
overlapped with PM pollution hotspots. Furthermore, as shown in Figure S3, the driving factors for urban
and rural PM pollution improvement differed significantly between land cover types.

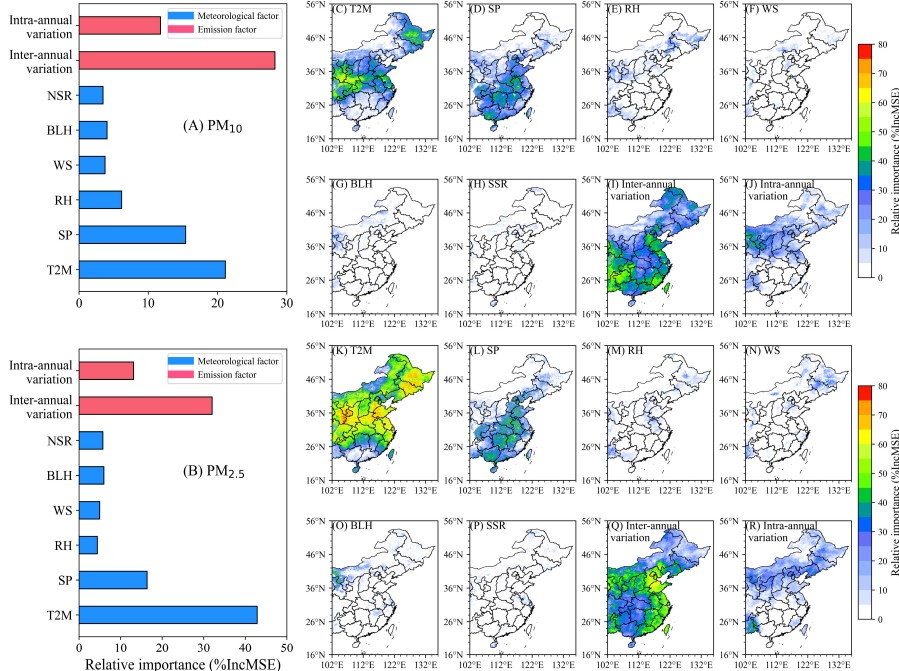


**Figure 5. Spatial distribution of the relative influence of each variable on PM pollution. In panels (A-B), the**
**red variables are related to emissions and the blue variables are related to meteorology.**

The relative contributions of each variable in the DOET model to the PM concentration values were
obtained using the permutation_importance library. The results showed that the improvement in urban
and rural PM pollution was primarily driven by interannual variation (Figure 5), followed by temperature,
which is consistent with the relative importance results in Figure 5. Figures S4-S5 illustrate how
variations in the values of the driving factors influence their relative contributions to PM concentrations.
In particular, PM concentrations showed a clear inverse relationship with temperature and interannual
variations, especially for PM$_{2.5}$. Relative humidity also showed clear differences in its contribution to
PM$_{10}$ and PM$_{2.5}$: lower relative humidity was associated with higher PM$_{10}$ concentrations, whereas higher
PM$_{2.5}$ concentrations were associated with higher relative humidity. The scatter plots illustrating the



relationships between other variables and their relative contributions to PM are shown in Figures S4-S5.

Figure 6 shows the relative contributions of each variable, with the spatial distribution patterns of

interannual variations being particularly noteworthy. For $PM_{10}$, regions such as Guanzhong, North China,
and Inner Mongolia were more susceptible to the influence of interannual variations. We hypothesize
that the improvement in $PM_{10}$ pollution be due not only be attributed to anthropogenic emission
reductions but also to sandstorm events in recent years, which are important sources of $PM_{10}$ (Wang et
al., 2024c). However, the explanatory power of the model for PM10 trends in these areas remains
relatively low, suggesting the need for further investigation into the specific causes. For $PM_{2.5}$, the impact
of interannual variability was observed mainly in the Guanzhong region, North China, and the Sichuan
Basin, all of which are key areas for pollution control (Wang et al., 2022a; Yu et al., 2022). Contrary to
the relative importance results, the dominant factor driving the improvement in urban and rural PM
pollution was the influence of interannual variability (Figure S6), with other variables showing varying
effects across different land cover types.

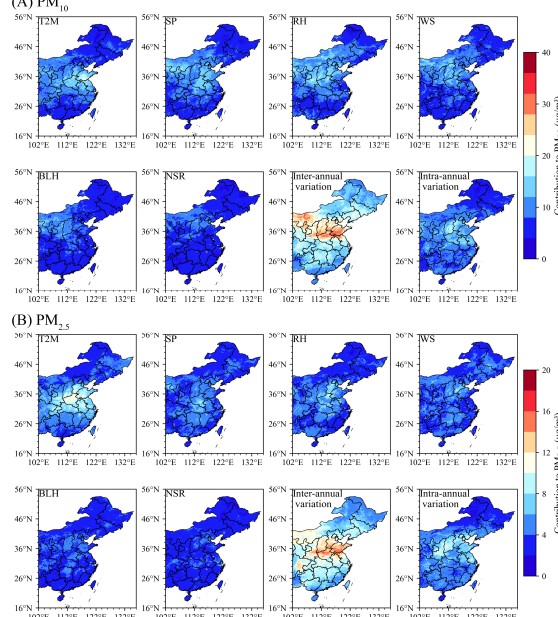

**Figure 6. The spatial distribution of the relative contributions of each variable to PM pollution**

Finally, the "tree_SHAP" tool was used to decompose the SHAP values of each variable in the

DOET model. By analyzing the positive and negative changes in the SHAP values, the influence of each
variable on the PM pollution improvement - whether positive or negative - was quantified, thus



complementing the assessment of driving factor contributions (Li et al., 2024a). As shown in Figure 7,
the SHAP values show a strong negative correlation between PM concentrations and the contribution of
interannual variability in eastern China. In particular, during the transition from 2019 to 2020, the
contribution of interannual variations to PM concentrations shifted critically from positive to negative.
Interestingly, despite the high relative importance and contribution of some variables, their SHAP values
showed periodic fluctuations, alternating between positive and negative, such as for temperature (with a
negative contribution in summer and a positive one in winter). This suggests that meteorological factors
influence PM concentrations in a periodic manner, while the only factor that consistently contributes to
the improvement of PM pollution is the interannual variation driven by anthropogenic influences.

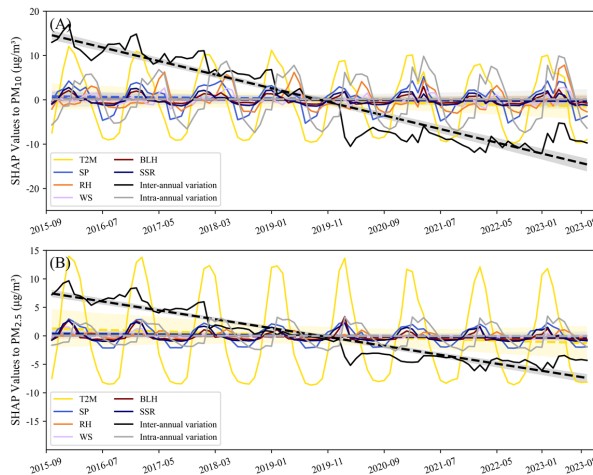


**Figure 7. The SHAP values of each variable for PM. The solid line represents the SHAP values, and the dashed line indicates their trend of change.**

**3.5 Trends in the contribution of driving factors to PM pollution improvement**

To further investigate the influence of potential driving factors on PM concentrations, we conducted
a detailed analysis of the trends in the contributions of each variable was performed. As shown in Figures
S7-S10, the monthly trends in the relative contributions and SHAP values of each variable were examined,
categorized into significant changes ($p < 0.05$) and non-significant changes ($p > 0.05$). For the relative
contributions (including $PM_{10}$ and $PM_{2.5}$), with the exception of interannual variations, all other variables
showed a decreasing trend, although some regions showed an increasing trend. However, the contribution
of interannual variability showed a significant decrease, indicating a reduced capacity of anthropogenic



emissions to trigger PM pollution events. This phenomenon is more pronounced for the trends in SHAP
values. In particular, only the contribution of interannual variations showed a significant decreasing trend,
while the other variables showed non-significant decreasing trends, mainly due to the periodic variations
in their contributions, as shown in Figure 7. This shows that the impact of a variable on PM pollution
cannot only be assessed on the basis of its relative contribution, but its positive or negative influence on
the improvement of PM pollution must also be considered.

Given the significant decrease in the contribution of interannual variation, we further compared its

trends across different land cover types in urban and rural areas, as this variable plays the most important
role in PM pollution improvement. As shown in Figure 8 (A-B), the trends in relative contributions for
both PM$_{10}$ and PM$_{2.5}$ did not differ significantly between the eight land cover types, although urban areas
showed the highest rate of decrease. However, the trends in SHAP values shown in Figures 8 (C-D)
revealed that the reduction in the contribution of interannual variation was most pronounced in urban
core areas, suburban areas, and towns. In contrast, the decrease in interannual contributions was more
pronounced in agricultural areas than in urban areas, while other rural areas showed a weaker influence
of interannual variations on PM pollution improvement. These results suggest that the improvement in
PM pollution in urban areas is more closely related to anthropogenic influences, whereas this relationship
is less pronounced in rural areas.

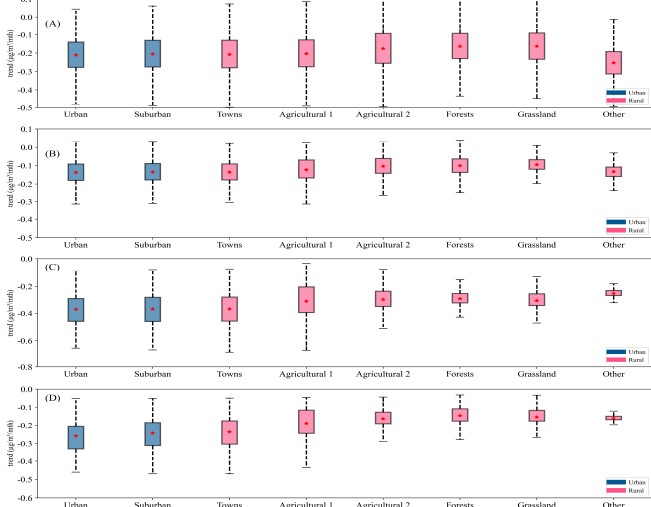

**Figure 8. Trends in the relative contribution (A-B) and SHAP values (C-D) of interannual variability of**
**different land cover types. A and C represent the case for PM$_{10}$, while B and D represent the case for PM$_{2.5}$.**
**In the legend, blue represents urban areas, and red represents rural areas.**





**4 Discussion and conclusion**

Due to the predominant distribution of environmental quality monitoring stations in urban areas (Park et al., 2020), discussions on air pollution patterns between urban and rural regions have been limited (Hammer et al., 2020). In this study, we used a regression-based machine learning DOET algorithm to integrate station-observed PM concentrations, satellite-observed TOAR, meteorological factors, and geographic information data. This approach enabled us to generate long-term, high spatio-temporal resolution datasets of near-surface $PM_{10}$ and $PM_{2.5}$, with a spatial resolution of 5 km, an hourly temporal resolution, and coverage across the entire eastern China region. Using the generated PM data in conjunction with a constructed urban-rural land type framework, we successfully captured the broad trends and patterns of $PM_{10}$ and $PM_{2.5}$ concentration changes from urban and suburban areas to different types of rural regions.

Based on the estimated dataset and interpretable parameters, the study identified significant large-scale improvements in PM pollution in eastern China from 2015 to 2023, indicating notable achievements from the implementation of clean air measures. The study noted that the second phase of the clean air action plan, implemented from 2018 to 2020, also produced positive results, following the success of the first phase from 2013 to 2017 (Geng et al., 2024). Our results show that under the urban-rural framework, PM reductions are generally higher in urban areas than in rural areas. However, the highly polluted agricultural areas in rural regions also showed significant improvements in PM pollution. In fact, during air pollution prevention and control efforts, China's main emission reduction measures focused on coal consumption and energy-intensive industries such as steel and cement, and these measures were often effective in urban areas (Yun et al., 2020; Huang et al., 2014b; Wang et al., 2013). This does not mean that rural areas have been neglected, as evidenced by reductions in biomass burning (Shen et al., 2019). The finding that interannual variability is the main driver of PM pollution improvement is consistent with these facts. It is worth noting that the rate of PM concentration decline is faster in urban areas than in rural areas, bringing the concentration levels of the two areas closer together. Given the more pronounced decrease in the contribution of inter-annual variations in urban areas, future efforts to prevent and control air pollution should maintain the current intensity or balance investments between urban and rural areas.

Our results indicate that meteorological factors with distinct seasonal variations, such as



temperature, boundary layer height, and relative humidity, have a cyclical influence on PM pollution.
For example, summer weather conditions, such as abundant precipitation, high relative humidity and
abundant water vapour favour PM dispersion, while winter weather conditions are less conducive to
pollutant dispersion and spring is often characterised by frequent dust events. Therefore, due to their
periodic positive and negative contributions and variability, meteorological conditions do not provide
stable improvements in PM pollution. Moreover, the contribution of meteorological conditions to PM
concentrations does not show a significant trend. Thus, given the high contribution of inter-annual
variability to the improvement of PM pollution, the impact of meteorological conditions on the inter-
annual variability of PM pollution in China should not be overemphasised.

Although this study evaluated the patterns of PM pollution improvement and its driving factors in

urban and rural areas of eastern China, the contribution of interannual variations driven by anthropogenic
influences was represented by a time variable in our analysis. In the future, key factors driving changes
in air pollutants, such as energy management, urban traffic management, agricultural nitrogen deposition
effects and biomass burning, need to be further incorporated into the attribution analysis to distinguish
and quantify the contributions of different anthropogenic emission reduction measures to PM pollution
improvement. Given the different drivers of PM pollution improvement in urban and rural areas, it is
essential to implement tailored strategies in both regions to achieve more effective and comprehensive
air pollution prevention and control measures in the future.
**Data availability**
The hourly ground station observations of near-surface $PM_{10}$ and $PM_{2.5}$ concentrations are obtained from
the China National Environmental Monitoring Center (CNEMC), which can be accessed on its official
website (http://www.cnemc.cn/en/). Himawari-8 TOAR data provided by the Japan Meteorological
Agency, download from: http://www.eorc.jaxa.jp/ptree/index.html. Meteorological variables were
derived from the reanalysis data set provided by the European Centre for Medium-Range Weather
Forecasts (ECMWF) (https://cds.climate.copernicus.eu/cdsapp#!/search?type=dataset). MODIS Land
use/cover        change        (LUCC)        product        can        be        downloaded        from
https://doi.org/10.5067/MODIS/MCD12C1.061. The 2015 UN-adjusted population density data (RK)
can be downloaded from https://doi.org/10.7927/H4PN93PB. SRTM-3 elevation data jointly measured



by NASA and the U.S. Department of Defense's National Imagery and Mapping Agency (NIMA)
(HEIGHT) can be downloaded from https://doi.org/10.5067/MEaSUREs/SRTM/SRTMGL3.003.

**Code availability**

The codes are available from the corresponding author upon request.

**Acknowledgements**

We would like to express our gratitude to the China National Environmental Monitoring Center,
Japan Meteorological Agency, European Centre for Medium-Range Weather Forecasts, and NASA
for their datasets.

**Financial support**

The work was supported by the Noncommunicable Chronic Diseases-National Science and Technology
Major Project (Grant number 2024ZD0531600), the National Natural Science Foundation of China
(Grant number 42427803), the Gansu Provincial Science and Technology Plan (Grant number
25RCKA024), and the Fundamental Research Funds for the Central Universities (Grant number lzujbky-
2023-ey10).

**Author contributions**

Z.S.: Software, Methodology, Data curation, Writing-Original draft preparation, Formal Analysis,
Visualization. B.C.: Conceptualization, Methodology, Writing-Reviewing and Editing, Resources.

**Competing interests**

The authors declare that they have no conflict of interest.

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
