# Peer review of "Urban-rural patterns and driving factors of particulate 2 matter pollution decrease in eastern china"

_EGUsphere, 2025_

## Author Comment (AC1)

Review result of "Urban-rural patterns and driving factors of particulate matter pollution decrease in eastern china" (egusphere-2025-2194)

Response to Reviewer #2:
reviewer's comments are given in blue,
our responses are given in deep red.

**Some of the content in the manuscript have been revised and updated.**

We would like to thank the editor and reviewers for carefully reading the manuscript and providing detailed and constructive comments, which have helped a lot in improving the manuscript. We quote each comment below, followed by our response.

This Manuscript uses an Extreme Trees based machine learning model to identify the drivers in changes of urban and rural PM in China. This is a good effort, and the authors demonstrate reasonable applicability of their approach.

The authors are very grateful to the reviewers for their comments. We thank them for taking the time to review this manuscript and for their valuable suggestions, which have significantly improved the academic quality of this manuscript.

Following are key points that need to be addressed:

1. Since the authors use spatial - temporal datasets, why was LSTM and Convolutional Neural Networks not applied?

Thank you very much for your valuable suggestions. The main reason why LSTM and convolutional neural network (CNN) were not adopted to process spatio-temporal data is that the selection of models should be closely combined with the research objectives and data characteristics. The core objective of this study is to identify the driving factors of PM changes (such as human factors, meteorological conditions, etc.). Extreme tree models have significant advantages in feature importance analysis and nonlinear relationship modeling. They can directly quantify the contribution of each variable to PM changes (such as including the importance of permutation features, relative contribution, and Shapley value), while LSTM and CNN are more suitable for capturing complex temporal or spatial patterns. However, the support for feature interpretation of LSTM and CNN are relatively weak and there is no direct python that can be applied. In addition, extreme tree models can handle high-dimensional mixed data without complex sequence modeling. We believe that LSTM and CNN have great potential in spatio-temporal modeling and plan to explore their applications in subsequent research, such as handling more complex spatio-temporal dependencies based on the LSTM-CNN hybrid model. In conclusion, we employed an extreme tree model to achieve a high degree of match between the model selection and the research objective. In the future, we will further optimize the method in combination with the suggestions of the reviewers.

2. Figure 7: Explain the physical justification for why SHAP values for temperature are

negative in summer and positive in winter? Would these change between urban versus rural areas? For example, biogenic emissions might increase in summer at higher temperatures increasing secondary organic aerosol formation. In winter, reducing temperatures might increase demand for residential heating. Further discussions are needed here.

Thank you very much for your valuable suggestions. The phenomenon whereby the SHAP values for temperature in Figure 7 are negative in summer and positive in winter primarily stems from the SHAP value calculation method and the way in which temperature influences PM concentration. Firstly, SHAP values are calculated based on how a specific variable, such as air temperature, increases or decreases the average PM concentration. In other words, the SHAP value for temperature indicates its positive or negative impact on PM, reflecting how much the model-output PM concentration deviates from the average PM concentration value in either direction. Furthermore, research findings from several sources suggest that, in summer, rising temperatures may reduce PM levels through two mechanisms: enhancing photochemical reactions that generate ozone ($O_3$), which consumes some precursors, and promoting atmospheric dispersion that dilutes pollutants. This results in negative SHAP values. Conversely, in winter, low temperatures primarily increase PM concentrations by boosting coal-fired emissions due to heating demands, as well as creating poor dispersion conditions caused by temperature inversions. This leads to positive SHAP values. The two figures below show the SHAP values of various variables for PM in urban and rural areas, respectively. The impact of various variables, including temperature, on PM is primarily evident in urban areas, where the magnitude of the values and the rate of change are both higher than in rural areas.

[Figure]

Figure S8. The SHAP values of each variable for PM in urban.

[Figure]

Figure S9. The SHAP values of each variable for PM in rural.

3. Figure 8 and related discussions: The figure is not clear. Discussions on lines 352-356 suggest different results for how interannual variations change between urban and rural areas using the 2 approaches: Relative contributions versus SHAP. Why are these different? Are SHAP values more reliable? The authors seem to just combine results from relative contribution and SHAP in their Abstract and Discussions. However, physical justification is needed to figure out what causes these differences.

Thank you very much for your suggestions. This study used relative contributions and SHAP values to explore the drivers of PM changes. However, the calculation methods for these two values differ. Relative contributions enable the results of each PM prediction to be broken down into bias and feature contributions. Each prediction can be presented as a simple sum of feature contributions, showing how features lead to a specific prediction. For a dataset with n features, each prediction is decomposed as follows: prediction = bias + feature_1_contribution + ... + feature_n_contribution. Here, 'bias' represents the model's inherent deviation, while 'feature_n_contribution' quantifies the magnitude of each variable's influence on the model output.

SHAP, based on the Shapley value from game theory, quantifies the positive or negative impact of each feature on the model prediction by calculating its average marginal contribution across all possible combinations of features. This method systematically eliminates interference from other features in order to assess the role of each feature in different combinations. Ultimately, it decomposes the model output into the independent contribution of each feature. For a dataset with n features, the SHAP value can be expressed as follows: prediction = mean + feature_1_contribution + ... + feature_n_contribution. Here, 'mean' is the average value of the PM time series and 'feature_n_contribution' is the feature SHAP value, representing the positive or negative impact of each variable on the model output and carrying a sign.

Thus, both relative contributions and SHAP values indicate the contribution of a variable within the model. The difference between them is that relative contributions break down each feature's contribution to a specific prediction, while SHAP values also show whether the feature's impact is positive or negative. As we clarified in the description of Figure 8, while the relative contribution suggests a significant contribution from interannual variability, the declining trend is not evident because it does not reveal that interannual variability has become a negative contributor to PM pollution. SHAP values, however, capture this distinction. The SHAP value changes for urban and rural areas shown in response to the second question demonstrate that the trend of lower SHAP values for interannual variability in urban regions is reasonable.

4. What about role of photochemistry? The authors include solar radiation, however, it does not show up as a key variable in SHAP interpretability analyses.

Thank you very much for your suggestion. First, we have corrected an error: we use the "net solar radiation at the surface" variable from ERA5 data to represent solar radiation, abbreviated as NSR in the text. However, in the previous version, this variable was incorrectly labeled as SSR in some figures. This has now been rectified. The current estimation model can be expressed as:

$$(PM_{10}, PM_{2.5}) = f\begin{pmatrix} TOAR_{1,2,3,4,6}, BLH, RH, SP, T2M, WD, WS, NSR, Height, LUCC, RK, \\ year, mon, doy, hour, lon, lat, SAA, SAZ, SOA, SOZ \end{pmatrix}$$

Then, in the SHAP interpretability analysis diagram (Figure 7), solar radiation is represented as NSR. Studies indicate that photochemical reactions play a significant role in the formation and transformation of fine particulate matter in the atmosphere through the "new particle formation effect" (Guo et al., 2020). Our SHAP analysis results also suggest that NSR influences PM and exhibits periodic variations.

5. Conclusions: Line 402-405: The authors rightfully acknowledge that anthropogenic influences are just represented by a time variable in their analyses. This is clearly insufficient. If possible, the authors should consider emissions, photochemistry (ozone, OH radicals, NOx, VOCs) etc. in their analyses.

Thank you very much for your suggestion. Time variables (year, month) effectively characterize cyclical patterns and long-term trends in human activity, serving as reliable proxy indicators in pollution analysis (Song et al., 2023). Monthly cycles directly reflect seasonal rhythms: winter heating spikes $PM_{2.5}$ and $SO_2$ levels (Liu et al., 2017), agricultural phases amplify ammonia emissions (Ma et al., 2025), and transportation peaks during holidays elevate $NO_2$ concentrations (Hua et al., 2021). Annual trends capture industrial evolution and policy impacts, such as the $PM_{2.5}$ reduction after implementing the "Air Pollution Prevention Action Plan" (Geng et al., 2024; Geng et al., 2021). As standardized, quantifiable metrics, time variables circumvent data limitations for complex activities (e.g., energy consumption, economic behaviors, urban sprawl), enable cross-regional comparisons without normalization, and reveal pollution

responses to socioeconomic rhythms and policy efficacy (Dai et al., 2021; Shi et al., 2021). Furthermore, due to data limitations, it is extremely challenging to fully account for emissions and photochemical parameters (such as ozone, hydroxyl radicals, NOx, and VOCs). Therefore, we employed a time variable to simply represent human influence while applying meteorological normalization to PM data to eliminate the impact of sudden meteorological events, thereby ensuring the validity of our data analysis. In the future, we will explore methods for obtaining long-term emission and photochemical data for analysis to enhance the related research. We have added relevant explanations in Section 2.3 of the Methods section of the manuscript to explain the rationality of using time variables (year, month) as proxies for human activity.

References:

Dai, Q., Hou, L., Liu, B., Zhang, Y., Song, C., Shi, Z., Hopke, P. K., and Feng, Y.: Spring Festival and COVID-19 Lockdown: Disentangling PM Sources in Major Chinese Cities, Geophysical Research Letters, 48, e2021GL093403, https://doi.org/10.1029/2021GL093403, 2021.

Geng, G., Liu, Y., Liu, Y., Liu, S., Cheng, J., Yan, L., Wu, N., Hu, H., Tong, D., Zheng, B., Yin, Z., He, K., and Zhang, Q.: Efficacy of China's clean air actions to tackle PM2.5 pollution between 2013 and 2020, Nature Geoscience, 17, 987-994, 10.1038/s41561-024-01540-z, 2024.

Geng, G., Xiao, Q., Liu, S., Liu, X., Cheng, J., Zheng, Y., Xue, T., Tong, D., Zheng, B., Peng, Y., Huang, X., He, K., and Zhang, Q.: Tracking Air Pollution in China: Near Real-Time PM2.5 Retrievals from Multisource Data Fusion, Environmental Science & Technology, 55, 12106-12115, 10.1021/acs.est.1c01863, 2021.

Guo, S., Hu, M., Peng, J., Wu, Z., Zamora, M. L., Shang, D., Du, Z., Zheng, J., Fang, X., Tang, R., Wu, Y., Zeng, L., Shuai, S., Zhang, W., Wang, Y., Ji, Y., Li, Y., Zhang, A. L., Wang, W., Zhang, F., Zhao, J., Gong, X., Wang, C., Molina, M. J., and Zhang, R.: Remarkable nucleation and growth of ultrafine particles from vehicular exhaust, Proceedings of the National Academy of Sciences, 117, 3427-3432, 10.1073/pnas.1916366117, 2020.

Hua, J., Zhang, Y., de Foy, B., Mei, X., Shang, J., and Feng, C.: Competing PM2.5 and NO2 holiday effects in the Beijing area vary locally due to differences in residential coal burning and traffic patterns, Science of The Total Environment, 750, 141575, https://doi.org/10.1016/j.scitotenv.2020.141575, 2021.

Liu, P., Zhang, C., Xue, C., Mu, Y., Liu, J., Zhang, Y., Tian, D., Ye, C., Zhang, H., and Guan, J.: The contribution of residential coal combustion to atmospheric PM2. 5 in northern China during winter, Atmos. Chem. Phys., 17, 11503-11520, 10.5194/acp-17-11503-2017, 2017.

Ma, S., Wang, N., Zhang, J., Ye, D., and Wang, L.: Ammonia chemistry and oxidation dynamics as dual driving factors of PM2.5 nitrate pollution: Insights from the spatiotemporal disparities in central China, Journal of Environmental Management, 392, 126594, https://doi.org/10.1016/j.jenvman.2025.126594, 2025.

Shi, Z., Song, C., Liu, B., Lu, G., Xu, J., Van Vu, T., Elliott, R. J. R., Li, W., Bloss, W. J., and Harrison, R. M.: Abrupt but smaller than expected changes in surface air quality attributable to

COVID-19 lockdowns, Science Advances, 7, eabd6696, 10.1126/sciadv.abd6696, 2021.

Song, C., Liu, B., Cheng, K., Cole, M. A., Dai, Q., Elliott, R. J. R., and Shi, Z.: Attribution of Air Quality Benefits to Clean Winter Heating Policies in China: Combining Machine Learning with Causal Inference, Environmental Science & Technology, 57, 17707-17717, 10.1021/acs.est.2c06800, 2023.

---

## Author Comment (AC2)

Review result of "Urban-rural patterns and driving factors of particulate matter pollution decrease in eastern china" (egusphere-2025-2194)

Response to Reviewer #1:
reviewer's comments are given in blue,
our responses are given in deep red.

**Some of the content in the manuscript have been revised and updated.**

We would like to thank the editor and reviewers for carefully reading the manuscript and providing detailed and constructive comments, which have helped a lot in improving the manuscript. We quote each comment below, followed by our response.

This study applies machine learning to estimate hourly $PM_{2.5}$ and $PM_{10}$ concentrations across eastern China using Himawari-8 satellite data, analyzing trends, influencing factors (2015–2023), and urban–rural disparities. The results are well presented. Below are comments and suggestions for improving the manuscript:

The authors are very grateful to the reviewers for their comments. We thank them for taking the time to review this manuscript and for their valuable suggestions, which have significantly improved the academic quality of this manuscript.

The particulate matter designations "PM2.5" and "PM10" should consistently use subscript formatting (i.e., $PM_{2.5}$ and $PM_{10}$) throughout the manuscript for scientific precision.

Thank you very much for your suggestion. We have carefully revised all subscripts for $PM_{2.5}$ and $PM_{10}$ in the manuscript.

Numerous previous studies have derived hourly surface PM concentrations from Himawari-8 observations in China (doi:10.5194/acp-21-7863-2021). These should be briefly summarized in the Introduction.

Similarly, the Extreme Trees model has been previously applied successfully for satellite-based $PM_{2.5}$ (doi:10.1038/s41467-023-43862-3; doi:10.1016/j.rse.2020.112136) and $PM_{10}$ (doi:10.1016/j.envint.2020.106290) estimation. A concise summary of these efforts should be added. In addition, a clear justification for selecting this particular model over other machine learning approaches is needed.

Thank you very much for your suggestion. We have carefully supplemented the relevant research, and the specific references added are listed in the introduction section of the manuscript:

"Currently, many studies have used machine learning models to obtain particulate matter concentration products and apply them to pollution assessment (Chen et al., 2019; Huang et al., 2021). Among these, extreme tree models and data from the Himawari-8 satellite have demonstrated outstanding performance (Wei et al., 2021b; Wei et al., 2021a; Wei et al., 2021c). In particular, the extreme tree model demonstrates its unique

advantages, including greater randomness and interference resistance, and outperforms other similar models in terms of performance (Wei et al., 2023)."

Line 91: The acronym "TOAR" appears before it is defined. All acronyms should be spelled out at first mention for clarity (e.g., "Tropospheric Ozone Assessment Report (TOAR)").
Thank you very much for your suggestion. We carefully checked the manuscript for such issues and have made the necessary corrections. However, to avoid overly long subheadings, we have added the first occurrence of TOAR to Line83-84:"First, by integrating Himawari-8/9 satellite top-of-atmosphere reflectance (TOAR) observation data, meteorological data, and geographic information".

Lines 97–101: The authors should clarify whether only Himawari-8 data were used, or whether Himawari-9 (which became operational in December 2022) was included in the 2022–2023 period. This is important for ensuring temporal consistency.
Thank you very much for your suggestion. After careful review, this paper utilizes data from Himawari-8 for the period from 2022 to 2023, and also includes data from the Himawari-9 satellite, which was launched during the same period. The time range for Himawari-8 data is from September 1, 2015, to September 30, 2022, while the time range for Himawari-9 data is from October 1, 2022, to August 31, 2023.

Lines 119–121: The data sources and preprocessing steps for elevation (HEIGHT), land use and land cover (LUCC), and population density (RK) should be explicitly described.
Thank you very much for your suggestion. We added relevant content about geographic information data to the manuscript. "HEIGHT is derived from SRTM-3 elevation data, with a spatial resolution of 90 meters and a temporal resolution of 1 year. The download URL is https://doi.org/10.5067/MEaSUREs/SRTM/SRTMGL3.003. LUCC is sourced from the dataset (MCD12Q1), with a spatial resolution of 500 meters and a temporal resolution of 1 year. The download URL is https://doi.org/10.5067/MODIS/MCD12Q1.006, used to describe land surface types and land use conditions. RK is derived from the 2015 United Nations adjusted population density data, with a spatial resolution of 0.1° × 0.1° and a temporal resolution of 1 year, available at https://doi.org/10.7927/H4PN93PB. It is provided by the Social and Economic Data and Applications Center (SEDAC) of the National Aeronautics and Space Administration (NASA)."

Equation 1: The model uses only top-of-atmosphere (TOA) reflectance, without accounting for viewing or solar illumination angles, which are known to influence aerosol retrievals. The authors should provide justification for their exclusion.
Thank you very much for your suggestion. We added observational geometric conditions (viewing Angle and solar altitude Angle) to the model to improve the estimation model. The formula of the current estimation model can be expressed as follows:

$$(PM_{10}, PM_{2.5}) = f \begin{pmatrix} TOAR_{1,2,3,4,6}, BLH, RH, SP, T2M, WD, WS, NSR, Height, LUCC, RK, \\ year, mon, doy, hour, lon, lat, SAA, SAZ, SOA, SOZ \end{pmatrix} \quad (1)$$

All the estimated and analytical data in the manuscript have been recalculated and plotted to ensure the accuracy of the analysis.

Figure 1: The methodology used to simultaneously estimate $PM_{2.5}$ and $PM_{10}$ via a multi-output model is unclear. A brief explanation or schematic would improve reader understanding.

Thank you very much for your suggestion. We added the principle of a multi-output model for simultaneously estimating $PM_{10}$ and $PM_{2.5}$ to the manuscript. The specific estimation process of the DOET model is as follows: firstly, meteorological factors, geographic information, and satellite TOAR data are input into the DOET model and matched with PM observation data. Then, the DOET model fits the PM observation data with the input variables to obtain two ET estimation models ($PM_{10}$ and $PM_{2.5}$). Finally, the two ET models are integrated to obtain the DOET model, and the estimation results of $PM_{10}$ and $PM_{2.5}$ are output simultaneously to save computation time. Finally, the obtained $PM_{10}$ and $PM_{2.5}$ data are subjected to further analysis.

Equation 2: The terms SS_res and SS_tot should be formally defined in the text or figure caption.

Thank you very much for your suggestion. In Section 2.3, we added detailed descriptions of the coefficient of determination ($R^2$), root mean square error (RMSE), and mean absolute error (MAE). In Equation (2), $ss_{res}$ represents the error between the estimated value of the model and the average value of the observed values of $PM_{10}$ and $PM_{2.5}$, $SS_{tot}$ represents the error between the observed values of $PM_{10}$ and $PM_{2.5}$ and the average value of the observed values of $PM_{10}$ and $PM_{2.5}$ from CNEMC. In Equation (3-5), $\hat{y}_i$ represents the $PM_{10}$ and $PM_{2.5}$ estimated value of the DOET model, $y_i$ represents the observed value of $PM_{10}$ and $PM_{2.5}$ from CNEMC.

Line 163: "SHAP" should be spelled out as "SHapley Additive exPlanations (SHAP)" upon first use.

Thank you very much for your suggestion. We have revised the relevant content in the manuscript.

Line 168: The selection of "20 times" for permutation testing appears arbitrary. A statistical or methodological justification is necessary.

Thank you very much for your suggestion. The statistical basis for the selection of "20 times" permutation test comes from the following reference (Qu et al., 2023). We referred to this article when using this method and the authors repeated the calculation of permutation importance 20 times to avoid uncertainty in the machine learning model, that we continued to use this method in our work.

Line 172: The purpose of the provided URL is unclear. The authors should clarify what resource it links to and its relevance.

Thank you very much for your suggestion. This URL link provides detailed information about the tree interpreter calculation method. The relevant details have been added to the manuscript.

Lines 212–213: The reported temporal cross-validation $R^2$ values (0.41 for $PM_{10}$, 0.51 for $PM_{2.5}$) seem inconsistent with the claim of "robust stability." The authors should address this discrepancy or revise the description accordingly.

Thank you very much for your suggestion. The expression in the manuscript has been revised to ensure the accuracy of the description of the research results: "The DOET model is relatively robust based on sample and spatial validation results".

Figure 2: The placement of accuracy labels is too close to the subplot boundaries, potentially affecting readability. Adjust the positions to improve visual clarity.

Thank you very much for your suggestion. The layout of Figure 2 has been Adjusted in the manuscript.

[Figure]

**Figure 2. Spatial distribution of PM₁₀ and PM₂.₅ and cross validation results of the DOET model. The dashed lines represent the 1:1 line, while the solid lines show the fitted line between observed and estimated values.**

Line 243: The manuscript does not evaluate relative reduction trends (i.e., trends normalized by baseline concentrations), which are crucial for comparing changes across regions with differing pollution levels. Consider incorporating this analysis.

Thank you very much for your suggestion. We supplemented our analysis by examining the relative change trends through benchmark concentration standardization. Initially, the standard deviation of PM concentrations was computed for each grid point to assess spatial variability. Subsequently, the annual mean PM data were used to calculate yearly relative changes normalized against benchmark concentrations. Finally, a comprehensive trend analysis was performed on these standardized values. The results are presented in Figure S2. Consistent with the overall trends in PM concentrations, the relative change rates of PM₂.₅ were quantified as −38.24 ± 3.40%/yr in rural areas and −40.93 ± 1.91%/yr in urban areas. Similarly, PM₁₀ exhibited relative change trends of −34.03 ± 6.55%/yr (rural) and −39.07 ± 2.78%/yr (urban). These findings demonstrate that, when accounting for region-specific baseline concentrations across different land cover types, urban areas continue to show a more substantial reduction in PM pollution compared to rural areas.

[Figure]

Figure S2. Analysis of PM concentration relative change trends in eastern China from September 2015 to August 2023.

Figure 3: Clearly define the boundaries (e.g., interquartile range, whiskers) of the box plots in the caption. Additionally, the color bar ranges in panels C–F are too broad, masking regional differences. Narrowing the ranges would better highlight spatial variability.

Thank you very much for your suggestion. The color bar ranges of Figure 3 (C-F) has been Adjusted in the manuscript. The boundaries of the box plot (such as interquartile range and whisker range) have been explained in detail and added to the description in Figure 3.

[Figure]

**Figure 3. Analysis of PM concentration trends in eastern China from September 2015 to August 2023. Panels A, C, D, and G represent PM$_{10}$, while panels B, E, F, and H represent PM$_{2.5}$. In the legends of panels G-H, blue indicates urban areas, and red indicates rural areas. In G and H, the upper part of the box represents the upper quartile of the trend, and the lower part represents the lower quartile of the trend; the dotted line range represents the upper and lower limits of the trend values; the red dot represents the average value of the trend.**

Lines 263–269: The number of decimal places reported is inconsistent. Standardize numerical precision across the section, preferably to two decimal places.

Thank you very much for your suggestion. We carefully checked the number of decimal places in all data in the manuscript to ensure that they were uniformly two decimal places.

Lines 277–278: The inclusion of temporal variables (year and month) as proxies for anthropogenic drivers requires further explanation. Clarify their interpretability in the context of human activity patterns.

Thank you very much for your suggestion. Time variables (year, month) effectively characterize cyclical patterns and long-term trends in human activity, serving as reliable proxy indicators in pollution analysis (Song et al., 2023). Monthly cycles directly reflect seasonal rhythms: winter heating spikes PM$_{2.5}$ and SO$_2$ levels (Liu et al., 2017), agricultural phases amplify ammonia emissions (Ma et al., 2025), and transportation peaks during holidays elevate NO$_2$ concentrations (Hua et al., 2021). Annual trends capture industrial evolution and policy impacts, such as the PM$_{2.5}$ reduction after implementing the "Air Pollution Prevention Action Plan" (Geng et al., 2024; Geng et al., 2021). As standardized, quantifiable metrics, time variables circumvent data limitations

for complex activities (e.g., energy consumption, economic behaviors, urban sprawl), enable cross-regional comparisons without normalization, and reveal pollution responses to socioeconomic rhythms and policy efficacy (Dai et al., 2021; Shi et al., 2021). Furthermore, due to data limitations, it is extremely challenging to fully account for emissions and photochemical parameters (such as ozone, hydroxyl radicals, NOx, and VOCs). Therefore, we employed a time variable to simply represent human influence while applying meteorological normalization to PM data to eliminate the impact of sudden meteorological events, thereby ensuring the validity of our data analysis. In the future, we will explore methods for obtaining long-term emission and photochemical data for analysis to enhance the related research. We have added relevant explanations in Section 2.3 of the Methods section of the manuscript to explain the rationality of using time variables (year, month) as proxies for human activity.

Figure 8A: The x-axis range is too narrow, truncating some boxplot distributions. Expanding the axis limits would allow for clearer visualization of data variability.
Thank you very much for your suggestion. The layout of Figure 8 (A) has been Adjusted in the manuscript. The boundaries of the box plot (such as interquartile range and whisker range) have been explained in detail and added to the description in Figure 8.

[Figure]

**Figure 8. Trends in the relative contribution (A-B) and SHAP values (C-D) of interannual variability of different land cover types. A and C represent the case for PM$_{10}$, while B and D represent the case for PM$_{2.5}$. In the legend, blue represents urban areas, and red represents rural areas. In Figure 8, the upper part of the box represents the upper quartile of the trend, and the lower part represents the lower quartile of the trend; the dotted line range represents the upper and lower limits of the trend values; the red dot represents the average value of the trend.**

References:

Dai, Q., Hou, L., Liu, B., Zhang, Y., Song, C., Shi, Z., Hopke, P. K., and Feng, Y.: Spring Festival and COVID-19 Lockdown: Disentangling PM Sources in Major Chinese Cities, Geophysical Research Letters, 48, e2021GL093403, https://doi.org/10.1029/2021GL093403, 2021.

Geng, G., Liu, Y., Liu, Y., Liu, S., Cheng, J., Yan, L., Wu, N., Hu, H., Tong, D., Zheng, B., Yin, Z., He, K., and Zhang, Q.: Efficacy of China's clean air actions to tackle PM2.5 pollution between 2013 and 2020, Nature Geoscience, 17, 987-994, 10.1038/s41561-024-01540-z, 2024.

Geng, G., Xiao, Q., Liu, S., Liu, X., Cheng, J., Zheng, Y., Xue, T., Tong, D., Zheng, B., Peng, Y., Huang, X., He, K., and Zhang, Q.: Tracking Air Pollution in China: Near Real-Time PM2.5 Retrievals from Multisource Data Fusion, Environmental Science & Technology, 55, 12106-12115, 10.1021/acs.est.1c01863, 2021.

Hua, J., Zhang, Y., de Foy, B., Mei, X., Shang, J., and Feng, C.: Competing PM2.5 and NO2 holiday effects in the Beijing area vary locally due to differences in residential coal burning and traffic patterns, Science of The Total Environment, 750, 141575, https://doi.org/10.1016/j.scitotenv.2020.141575, 2021.

Liu, P., Zhang, C., Xue, C., Mu, Y., Liu, J., Zhang, Y., Tian, D., Ye, C., Zhang, H., and Guan, J.: The contribution of residential coal combustion to atmospheric PM2.5 in northern China during winter, Atmos. Chem. Phys., 17, 11503-11520, 10.5194/acp-17-11503-2017, 2017.

Ma, S., Wang, N., Zhang, J., Ye, D., and Wang, L.: Ammonia chemistry and oxidation dynamics as dual driving factors of PM2.5 nitrate pollution: Insights from the spatiotemporal disparities in central China, Journal of Environmental Management, 392, 126594, https://doi.org/10.1016/j.jenvman.2025.126594, 2025.

Qu, S., Liu, J., Li, B., Zhao, L., Li, X., Zhang, Z., Yuan, M., Niu, Z., and Lin, A.: Unveiling the driver behind China's greening trend: urban vs. rural areas, Environmental Research Letters, 18, 084027, 10.1088/1748-9326/ace83d, 2023.

Shi, Z., Song, C., Liu, B., Lu, G., Xu, J., Van Vu, T., Elliott, R. J. R., Li, W., Bloss, W. J., and Harrison, R. M.: Abrupt but smaller than expected changes in surface air quality attributable to COVID-19 lockdowns, Science Advances, 7, eabd6696, 10.1126/sciadv.abd6696, 2021.

Song, C., Liu, B., Cheng, K., Cole, M. A., Dai, Q., Elliott, R. J. R., and Shi, Z.: Attribution of Air Quality Benefits to Clean Winter Heating Policies in China: Combining Machine Learning with Causal Inference, Environmental Science & Technology, 57, 17707-17717, 10.1021/acs.est.2c06800, 2023.

---

## Author Response (AR2)

Review result of "Urban-rural patterns and driving factors of particulate matter pollution decrease in eastern china" (egusphere-2025-2194)

Response to Editor:
editor's comments are given in blue,
our responses are given in deep red.

**Some of the content in the manuscript have been revised and updated.**

We would like to thank the editor for carefully reading the manuscript and providing detailed and constructive comments, which have helped a lot in improving the manuscript. We quote each comment below, followed by our response.

Please notice that the journal now have author guidelines for Title, Abstract, and Conclusions (https://www.atmospheric-chemistry-and-physics.net/policies/guidelines_for_authors.html). I found the following is missing "The topic of the article and why it is important" in the abstract. Please follow the guidelines to make justifications.

Thank you very much for your valuable suggestions. When we previously drafted the abstract of this manuscript, we did not clearly articulate the section on 'The topic of the article and why it is important'. This study's primary objective is to integrate an interpretable artificial intelligence framework with multi-source data (satellite, meteorological and auxiliary data) to derive particulate matter concentrations in eastern China over the past several years, assessing trends in urban and rural areas. Interpretable variables are then used to analyses the model and identify the key variables influencing pollutant variation. This research is crucial for evaluating the effectiveness of environmental policies and ensuring equitable health co-benefits. Based on the above, we have revised the opening section of the abstract to include the part about 'The topic of the article and why it is important'. The specific content is as follows: '[*Understanding the urban-rural patterns and driving drivers behind the recent decrease in particulate matter (PM) pollution across eastern China is essential for assessing the efficacy of environmental policies and ensuring equitable health co-benefits. By employing an interpretable, end-to-end machine learning framework integrating satellite observations, meteorological factors, and auxiliary datasets, this study reveals changes in urban and rural PM pollution and the underlying drivers.*]', and the corresponding modifications have been made in the manuscript.